# Analysis of Volatile Flavor Substances in the Enzymatic Hydrolysate of *Lanmaoa asiatica* Mushroom and Its Maillard Reaction Products Based on E-Nose and GC-IMS

**DOI:** 10.3390/foods11244056

**Published:** 2022-12-15

**Authors:** Ning Yang, Shasha Zhang, Pei Zhou, Weisi Zhang, Xiaoli Luo, Jingjing Cao, Dafeng Sun

**Affiliations:** 1Yunnan Academy of Edible Fungi Industry Development, Kunming 650221, China; 2Kunming Edible Fungi Institute of All China Federation of Supply and Marketing Cooperatives, Kunming 650221, China

**Keywords:** *Lanmaoa asiatica*, enzymatic hydrolysate, Maillard reaction, E-Nose, GC-IMS, volatile-flavor compounds (VFCs)

## Abstract

An electronic nose (E-Nose) and gas chromatography-ion mobility spectrometry (GC-IMS) were used to analyze the volatile flavor compounds (VFCs) of the enzymatic hydrolysate of *Lanmaoa asiatica* and its Maillard reaction products (MRPs). E-Nose sensors have strong response signals to sulfide, nitrogen oxides, alcohols, and aldehyde ketone, and the aroma profile was increased after the Maillard reaction (MR). According to GC-IMS, A total of 84 known compounds were identified. Aldehydes, ketones and alcohols are the main VFCs. After MR, the concentrations of some alcohols decreased, and the concentration of pyrazines and ketones increased. Principal component analysis (PCA) and similarity analysis showed that the enzymatic hydrolysate and MRPs were different and could be effectively distinguished. In conclusion, this study clarified the changes in VFCs before and after the MR. The results can provide a theoretical basis for the quality control and flavor changes during the processing of *Lanmaoa asiatica* and provide a new method for flavor analysis of edible mushrooms and their products.

## 1. Introduction

*Lanmaoa asiatica (L. asiatica)* is a famous wild edible mushroom widely distributed in Yunnan, China, and belongs to the Boletaceae family [1]. Similar to other bolete mushrooms, *L. asiatica* is popular in its origin country for its flavorful smell and delicious crunchy taste [2]. *L. asiatica* is rich in protein, crude fiber, minerals and other nutrients. In addition, it contains polysaccharides, terpenes and other active substances, which can effectively play antioxidant, antibacterial, and regulate the immunity of the human body, and has high edible and medicinal value [3]. However, it is yet to be cultivated artificially, and its fresh fruitbody will decay rapidly under cold storage conditions. Further processing of *L. asiatica* is one of the solutions to the above problem, which can utilize the *L. asiatica* resource more efficiently and increase its commercial value. However, to our knowledge, the changing profiles of volatile compounds during processing are less reported.

Flavor is one of the most important qualities of food, and flavor changes can affect the sensory characteristics and influence consumer preferences [4]. In recent years, the aroma and taste of bolete mushrooms have received a lot of attention from researchers. Some research shows that bolete mushrooms contain about 175 volatile compounds, including octacarbons, aldehydes, esters, etc.; 1-octen-3-ol, 3-octanone and 3-octanol are the main characteristic aroma components [5]. However, there are differences in the aroma compounds of *Boletus* from different species and processing methods. We found 31 volatile compounds detected in fresh *L. asiatica* in our previous study [6]. Another study reported a total of 98 volatile compounds in dried *L. asiatica*, which was more abundant than fresh *L. asiatica* [7].

Enzymolysis is an important technique for extracting flavor substances from edible materials, however, enzymatic hydrolysates generally do not have prominent characteristic flavors. On the contrary, negative tastes such as bitterness are often seen. Maillard reaction (MR) is an effective means of flavor enhancement, which can improve food flavor and food stability [8,9]. Several studies have shown that the MR can eliminate the bad flavor of edible mushroom enzymatic hydrolysate and increase the flavor and umami [8,10,11]. Most of the studies only evaluated the sensory characteristics and the changes in non-volatile compounds content changes. Meanwhile, little information is available on the change in volatile flavor compounds (VFCs).

An electronic nose (E-Nose) is an efficient instrument for the rapid detection of food flavor and provides the overall VFC information of the tested samples [12,13]. It can compare and analyze the VFCs of between samples and overcome the problems of subjective influence and poor the repeatability of a human smeller. However, the E-Nose cannot provide the qualitative results of each substance/compound in the sample. Gas chromatography-ion mobility spectrometry (GC-IMS) is a technique that combines gas chromatography with ion mobility spectrometry to analyze both flavor compound and sample quality [14,15]. It characterizes the flavor differences between samples through a visual Gallery plot [5,15]. Currently, GC-IMS technology is widely used in food flavor testing. Based on the advantages and disadvantages, combining the E-Nose and GC-IMS will provide a more comprehensive result, as an E-Nose can quickly distinguish the aroma profile of a sample, and GC-IMS provides a richer data set for the qualitative analysis of each compound in the sample.

In this study, the enzymatic hydrolysate of *L. asiatica* and its Maillard reaction products (MRPs) were analyzed based on E-Nose and GC-IMS techniques. Differential analysis of the volatile components profile of two samples was investigated.

## 2. Materials and Methods

### 2.1. Materials and Equipment

The materials used in the study included: dried *L. asiatica* fruitbody, purchased from Kunming Mushuihua Wild Mushroom Trading Market; 5 × 10^4^ U/g neutral protease, 3 × 10^4^ U/g papain, purchased from Nanning Pangbo Bioengineering Co., Ltd. (Nanning, China); 5 × 10^4^ U/g flavor protease, purchased from Cangzhou Xiasheng Enzyme Biotechnology Co., Ltd. (Cangzhou, China).

The equipment used in the study included: EasyPlus Titrator ET18 automatic potentiometric titrator, Mettler-Toledo Instruments Ltd. (Shanghai, China); FiveEasy Plus FE28 pH meter, Mettler-Toledo Instruments Ltd.(Shanghai, China); HH-6 constant temperature water bath, Shanghai Lichen Technology Co., Ltd. (Shanghai, China); centrifuge, Shanghai Anting Scientific Instrument Factory (Shanghai, China); TYM-30L ultra-micro pulverizer, Jinan Ltd. (Jinan, China); PEN3 electronic nose, AIRSENSE Analytics GmbH (Schwerin, Germany); FlavourSpec^®^ flavor analyzer, G.A.S, Scientific Instruments Co., Ltd. (Dortmund, Germany).

### 2.2. Method

#### 2.2.1. Preparation of Enzymatic Hydrolysate of *L. asiatica*

The powder of *L. asiatica* was mixed with water at 1:20 (*w*/*v*) and supplemented with 0.6% neutropase, 0.9% (*w*/*w*) flavor protease, and 4.7% (*w*/*w*) papain, at pH 7.0. The reaction was maintained at 55 °C for 1.5 h. After the hydrolysis, the enzyme was inactivated in a boiling water bath for 10 min, and centrifugation (4000 rpm, 15 min) was performed at 60 °C The supernatant was separated to obtain the enzymatic hydrolysate of *L. asiatica*.

#### 2.2.2. Preparation of the Maillard Reaction Products

A certain amount of the *L. asiatica* enzymatic hydrolysate was weighed, and 10% (*w*/*w*) of fructose and 1.5% (*w*/*w*) of L-glutamic acid were added according to the parameters and conditions of the Maillard reaction determined in the previous research, the reaction time was 80 min, the reaction temperature was 120 °C, and the initial pH was 8.0. The Maillard reaction product was obtained after cooling. The product was stored at −80 °C for further analysis.

#### 2.2.3. E-Nose Analysis

Reference to and adjust the detection methods of Luo Ying et al. [16]. Aroma collection: take 10 mL of the sample (enzyme hydrolysate/MRPs) in the sample bottle at 60 °C of thermal insulation for 10 min; each sample is repeated three times. Detection conditions: use a single sample by manual sample injection, sensor cleaning time 60 s, sample preparation time 5 s, sample detection time 100 s, and sample inflow flow rate 400 mL/min. The sensor response characteristics are shown in Table 1.

#### 2.2.4. GC-IMS Analysis

Take 2 g of the sample and place it in a 20 mL headspace vial and incubate at 60 °C for 20 min. GC conditions: column wax (30 m, ID: 0.53 mm), film thickness 1 μm (RESTEK, Bellefonte, PA, USA), column temperature 60 °C; carrier gas/drift gas N_2_; initial gas flow rate 2.0 mL/min, hold for 2 min; then ramp up to 10 mL/min within 2~10 min; increase to 10 mL/min within 2~10 min. IMS conditions: temperature 45 °C; drift gas flow rate of 150 mL/min.

### 2.3. Data Analysis

E-Nose data analysis: Using the software Winmuster (version 1.6.2.18/ Sep 25, AIRSENSE Analytics GmbH, Schwerin, Germany) that comes with the PEN3 E-nose and excel (version 11.1.0.991, Beijing Kingsoft Office Software Co., Ltd., Beijing, China). The response value data from 57–60 s of the selected samples were processed and analyzed (each second was used as a separate value for a total of 4 s, i.e., 4 values). The analysis included a radar plot of the sensor values, Principal component analysis (PCA) and Loading analysis (LA) analysis.

GC-IMS data analysis: The analysis software accompanying the FlavourSpec^®^ flavour VOCal. (version 0.1.0, G.A.S. Gesellschaft für analytische Sensorsysteme mbH BioMedizin Zentrum Dortmund, Dortmund, Germany) was used to carry out the differential analysis of the profiles and qualitative analysis of the compounds through the IMS database and the NIST database built into the application software.

## 3. Results and Discussion

### 3.1. E-Nose Analysis

#### 3.1.1. Aroma Composition Analysis of the Enzyme Hydrolysate of *L. asiatica* and Its MRPs

To intuitively understand the main aroma of the volatile substances in the sample, we developed a radar plot of the E-Nose sensor response signals. The response signals of the E-Nose sensor to the volatile substances in the sample are shown in Figure 1. As can be seen from Figure 1, the sensors W1W, W2W, W5S, and W1S response signals were relatively strong in the samples. The volatiles in the samples mainly contain sulfide, alcohols, aldodes, nitrogen oxides, and methyl compounds. However, the flavor profile is increased after the Maillard reaction (MR), especially the sensor W5S response signal was significantly enhanced, indicating that the characteristic aroma represented by nitrogen oxides increased after the MR. Z Li et al. [17]. used an E-Nose to analyze the volatile substances of the fungus-bone (Sarcodon imbricatus and chicken bone) enzyme hydrolysate before and after the MR, and also found that the flavor substances were mainly sulfide, nitrogen oxides and alcohols, and the aroma profile was significantly increased after the MR. This is consistent with our finding that the MR is able to alter the aroma profile in the sample flavor.

The sensor differential contribution rate analysis (Loading analysis, LA) is a measure of the magnitude of the sensor contribution in the process of discrimination. It can confirm the contribution rate of each sensor to the sample differentiation, thus determining the aroma components that play the main role in the sample differentiation process. The results of the Loading analysis are shown in Figure 2. From Figure 2, the contribution rate of the first principal component (PC-1) was 96.52%, the second principal component (PC-2) was 3.01, and the total contribution was 99.53%. It showed that the first and second principal components could characterize most of the characteristic aromas in the samples. W1W showed the highest contribution rate on PC-1 and W5S on PC-2, indicating that sulfide was correlated highly with the PC-1, and nitrogen oxides correlated most strongly with the PC-2.

#### 3.1.2. PCA of the Enzyme Hydrolysate of *L. asiatica* and Its MRPs

Principal component analysis (PCA) is a commonly used chemometric method to reveal the relationship between variables through data dimensionality reduction [18]. As shown in Figure 3, the contribution rate of the first principal component (PC-1) was 96.52%. The second principal component (PC-2) was 3.01, and the first principal component played a major role in the sample. The enzyme hydrolysate samples are mainly distributed between 1.0 and 1.5, while the MRPs are mainly distributed between 2.5 and 3.5. The two samples can be well distinguished, indicating the flavor difference between the two.

### 3.2. GC-IMS Analysis

#### 3.2.1. GC-IMS Spectrum Analysis of the Enzymatic Hydrolysate of *L. asiatica* and Its MRPs

To directly compare the differences in volatile compounds in the samples, we created a two-dimensional top view and a difference spectrum (Figure 4). Figure 4A,B represent the top view and difference spectra of the two samples, respectively. The vertical coordinate represents the retention time (s) of the GC, the horizontal coordinate represents the ion migration time (normalized), the red vertical line at the horizontal coordinate 1.0 is the RIP peak (reactive ion peak, normalized), and each point on either side of the RIP peak represents a volatile compound [18]. As can be seen in Figure 1, the two samples have similar types of volatile compounds, but the signal intensity differs significantly.

The white dots in Figure 4A indicate that the concentration of the compounds is low, while the red dots indicate that the concentration of the compounds is high, and the color shade is directly proportional to the concentration. Figure 4A reflects the differences in the types and concentrations of compounds between the enzymatic hydrolysate and the MRPs. The migration times of both samples were within 1.0–1.5, but the retention times differed. The retention time of most compounds in the enzymatic hydrolysate was within 0–500 s, while the retention time of compounds in the MRPs was concentrated within 0–1000 s. In addition, the intensity of the compound signals in the MRPs changed. This indicates that the variety of compounds is more abundant, and the concentration of compounds is changed after the MR.

Enzymatic hydrolysate (E-1) was used as a control in Figure 4B. The remaining spectra were obtained by subtracting the signal peaks in E-1 to obtain the difference spectra. The blue dots in the graph indicate that the compound is in a sample with a lower concentration compared to the control sample. The red dots indicate that the compound is present in a sample with a higher concentration compared to the control sample. The darker the color, the greater the difference between the samples. The background is white after subtraction of the reference signal peak, indicating consistent compounds. The blue region indicates that the compound has a low signal intensity in the MRPs. The red region indicates that the compound has a high signal intensity in the MRPs. After the MR, the signal of some compounds decreased, indicating that the concentration of some compounds decreased during the MR, probably due to the instability of some compounds and their easy decomposition in a high-temperature environment. However, there is also an increase in the signal of some compounds, indicating that the MR can produce new compounds or increase the concentration of the original compounds.

#### 3.2.2. Differences in Volatile Compounds between the Enzymatic Hydrolysate of *L. asiatica* and Its MRPs

To visually and quantitatively compare the differences in the volatile compounds between the samples, we established a visual topography (Figure 5) and a fingerprint map (Figure 6). In Figure 5, a number represents a compound, and the color generation indicates the compound concentration. In Figure 6, each row represents all the signal peaks selected in one sample, and each column represents the signal peaks of the same volatile compound in different samples. “M” and “D” indicate monomers and dimers of the same compound, and the numbered peaks indicate unidentified peaks.

The topography (Figure 5) differentiates the volatile compounds in the samples by comparing the retention times and ion migration times. The fingerprint profile (Figure 6) provides a visual representation of the complete volatile compound information for each sample and the volatile compound differences between samples. As can be seen from Figure 2, the topography of the two samples is relatively similar, but with differences in volatile compound concentrations. Combining the analysis of Figure 5 and Figure 6, we found that 3-Methyl-1-butanol-D (22), Acetic acid (6), 3-Methyl-1-butanol-m (21), Methyl acetate (54), and other compounds in higher concentrations, in the enzymatic digest (E-1, E-2, E-3). In contrast, the concentrations of Pyrazine (65), 1-Hydroxy-2-propanone-D (82), 2,5-Dimethylpyrazine-m (70), Furfural (77), 3-Methyl-2-butanol (63), Methional (9), and other compounds in the MRPs (M-1, M-2, and M-3) was higher than that of the enzymatic hydrolysate.

After the MR, the concentration of some compounds decreases (shown in the red area of the graph) and the concentration of some compounds increases (shown in the green area of the graph). Changes in the type and concentration of volatile compounds can bring about changes in flavor. For example, increased pyrazine compounds can bring strong roasted meat and nut aromas, while esters and ketones can bring floral and fruit aromas [19]. In conclusion, the topographic map and fingerprinting showed that the volatile compounds of the enzymatic hydrolysate and the MRPs were significantly different, indicating that the MR was able to change the volatile compounds of the enzymatic hydrolysate and lead to flavor differences between the samples.

#### 3.2.3. GC-IMS Integral Parameter Analysis of Volatile Fractions in the Enzymatic Hydrolysate of *L. asiatica* and Its MRPs

In order to better exhibit the information on the volatile compounds in the samples, including the retention index and migration time data. We listed the retention index and migration time, and peak intensity data of all compounds in Table 2. The compounds listed in the table are those identified by comparison with the database. Nineteen compounds were not listed because they could not be identified. As shown in Table 1, a total of 84 known volatile compounds were detected by GC-IMS, including 19 aldehydes, 18 alcohols, 16 ketones, nine esters, eight pyrazines, four acids, three furans and seven others. Among them, five octa-carbon compounds were detected, including Phenylacetaldehyde, Octanal, 1-octen-3-ol, 6-methyl-5-hepten-2-one, and acetophenone. One study reported that 34 volatile compounds were detected in fresh *L. asiatica*, with aldehydes and alcohols being the main volatile compounds, while 98 volatile compounds were found in dried *L. asiatica*, with hydrocarbons, esters, and alcohols being the majority [6,7]. The number of compounds and the main flavor compounds detected in this study differed from those of fresh and dried *L. asiatica*. It is suggested that enzymatic and MR can affect the volatile compound variety. In particular, other flavor compounds such as aldehydes, pyrazines and ketones are easily produced in the process of the MR [20], which will change the original aroma characteristics of *L. asiatica*.

Aldehydes are mainly produced through lipid oxidation and MR. After MR, aldehydes are formed mainly through the decarboxylation and deamination reactions of amino acids [21]. Aldehydes have strong aromatization ability and low threshold, presenting fruit, fat and nut aromas [22,23,24]. The intensity of the peaks of phenylethylaldehyde, octanal, 2-methyl-2-pentenal, propanal, and furfural increased significantly after the MR.Phenylacetaldehyde and octanal are eight-carbon compounds with almond and floral aromas, which are the key aroma compounds in edible mushrooms [25].

Alcohol compounds are produced by the oxidative decomposition of oils and fats. However, alcohols have high threshold values and contribute little to food flavor [24]. After the MR, the intensity of most of the alcohol peaks in the samples weakened, indicating that alcohols are easily lost during the MR. Misharina et al. [26]. found that the concentration of alcohols in *Lentinula edodes* mushroom also decreased significantly during drying, which is consistent with our findings. Among them, 1-octen-3-ol, known as “mushroom alcohol”, has a mushroom aroma and is a key contributor to the special flavor of mushrooms [27]. However, 1-octen-3-ol contains unsaturated double bonds, which are chemically unstable and easily decomposed during processing [5]. One study reported that 1-octen-3-ol was not detectable in dried porcini mushrooms and that 1-octen-3-ol is easily decomposed in high-temperature and low-humidity environments [28]. Our results also revealed that the intensity of the 1-octen-3-ol peak decreases after the MR, which leads to a weakening of the mushroom aroma of the MR.

Ketones are the products of the oxidation of alcohols or the decomposition of esters, with floral and fruity aromas, their odor threshold concentration is low and has a significant improvement on food flavor [18]. The intensity of the ketone peaks changed significantly (enhanced or weakened) after the MR, and there were 12 compounds with enhanced peak intensity, including acetophenone, 1-hydroxy-2-propanone, 2,3-pentanedione, etc.

Pyrazines are also a class of compounds of interest. These compounds are easily produced in low water activity and high-temperature environments and are important products of the MR [29,30]. The intensity of most of the pyrazine peaks in the samples increased after the MR. One study reported that pyrazines were not detected in fresh *L. asiatica*, but two pyrazines were found in dried products [6,7]. It is suggested that the drying process is accompanied by the occurrence of MR, which produces pyrazine compounds capable of imparting a special grilled and saucy flavor to bolete mushrooms.

Three of the five detected octa-carbon compounds showed a significant increase in peak intensity after the MR. The increased concentrations of octa-carbons enhances the mushroom aroma and imparts a more intense characteristic flavor to the MRPs. In conclusion, the MR is able to change the concentration of volatile compounds in the enzymatic hydrolysate, which in turn leads to differences in flavor between the enzymatic hydrolysate and the MRPs.

### 3.3. Principal Component Analysis and Similarity Analysis of the Enzymatic Hydrolysate of L. asiatica and Its MRPs

PCA analysis was used to visualize the difference, similarity, and principal component contribution of the volatile compounds in the two samples by signal intensity. Figure 7A represents the PCA analysis graph of the sample. From the figure, the first principal component (PC 1) contributes 94%, the second principal component (PC 2) contributes 3%, and the total contribution is 97%. This indicates that there is a certain similarity between the two samples, mainly in the form of small differences in the composition of volatile compounds. In addition, the PCA results showed that the corresponding scatters within the group of samples of the enzymatic hydrolysate and its MRPs clustered with each other, indicating the high similarity of the samples within the group. While the two samples were distributed in different intervals, the enzymatic hydrolysate sample was distributed on the left side of the figure, and the MRPs were distributed on the right side of the figure, which indicating a good differentiation between the samples. This further indicated that the flavor difference between the enzymatic hydrolysate and the MRPs was significant. The results were consistent with the E-Nose PCA analysis, which further verified that the enzyme hydrolysate varies in different flavors from its MRPs.

For “Nearest Neighbor” fingerprinting, a quick comparison of the samples based on the intensity of the compounds in the selected evaluation region and the calculation of the Euclidean distance between every two samples was undertaken. By calculating and comparing Euclidean distances, which can be used for volatile chemosynthetic similarity analysis [30]. Figure 7 represents the nearest-neighbor-Euclidean distance diagram of the enzymatic hydrolysate and its MRPs. A closer distance indicates higher sample similarity; farther distance indicates higher sample variability. As shown in Figure 5, the two samples were far apart, indicating a large variability in volatile compounds between the samples, which was consistent with the results of PCA. It indicated that the volatile compounds of both the enzymatic hydrolysate and the MRPs changed significantly.

The results of PCA and nearest neighbor analysis showed that after the MR, there was less difference in the type of volatile compounds compared to the enzymatic hydrolysate, but the volatile compounds changed significantly, resulting in high differentiation between samples and significant flavor differences.

## 4. Conclusions

In this study, the E-Nose and GC-IMS technique was used to analyze the volatile flavor substances of the enzymatic hydrolysate of *L. asiatica* and its MRPs. The E-Nose results showed that the aroma profile changed after MR. A total of 84 volatile compounds were identified. After the MR, the concentration of most pyrazines and ketones increased, and the concentration of three octa-carbon compounds, Phenylacetaldehyde, Octanal, and Acetophenone, also increased. The enzymatic hydrolysate was similar to the MRPs in terms of the types of volatile compounds, but the concentrations differed significantly. Moreover, the PCA and nearest neighbor analysis showed that the enzymatic hydrolysate was highly differentiated from the MRPs samples, with significant changes in volatile compounds and large differences in flavor between the two. In summary, this study used the volatile substances of E-Nose discrimination enzyme hydrolysate and its MRPs, and combination with GC-IMS specific analysis of volatile differences. This study confirms that E-Nose combination with GC-IMS is a reliable technique for the analysis of VFCs and can efficiently identify volatile substances and flavor variations in processed products of *L. asiatica*. The results can lay the foundation for the product development of *L. asiatica*. In particular, it provides a theoretical basis for quality control and flavor changes during the processing of *L. asiatica*. However, the mechanism of the changes in the types and concentrations of volatile compounds after the MR is still unknown, and this needs further study. In addition, the reducing sugars and amino acid interactions in the enzymatic hydrolysate also produce the MRPs, but their effects on the flavor of the enzymatic hydrolysate also need to be further researched in order to provide a more theoretical basis to guide the product development of *L. asiatica*.

## Figures and Tables

**Figure 1 foods-11-04056-f001:**
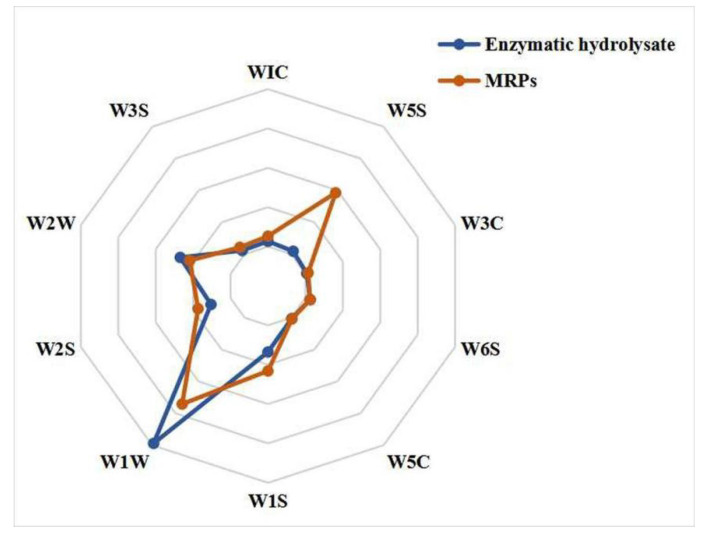
Sensor response signal radar plot.

**Figure 2 foods-11-04056-f002:**
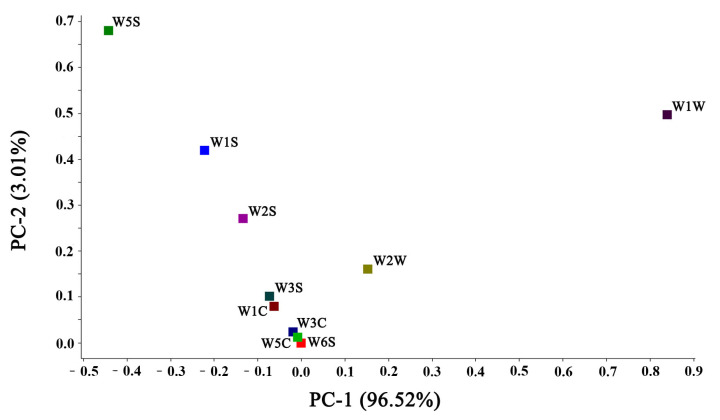
Results of the Loading analysis.

**Figure 3 foods-11-04056-f003:**
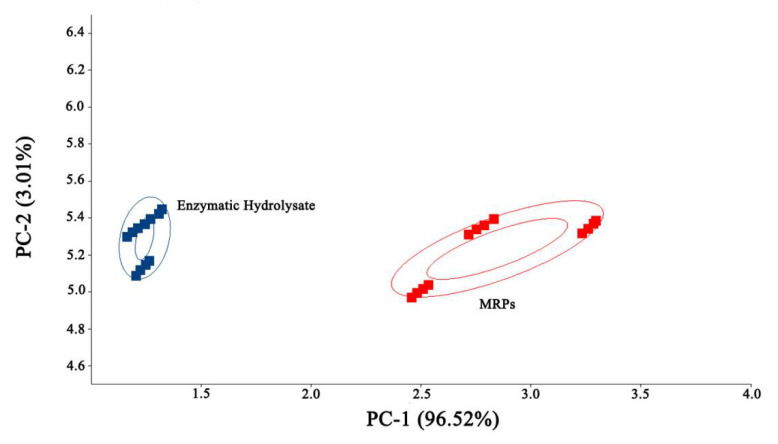
E-Nose PCA plot.

**Figure 4 foods-11-04056-f004:**
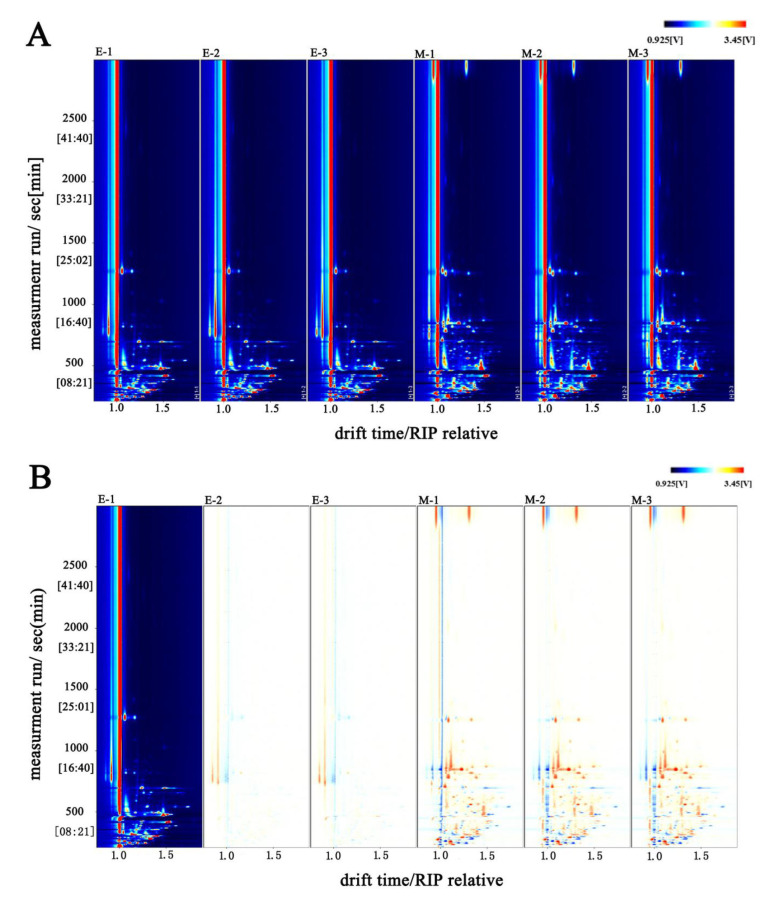
GC-IMS spectrum of enzymatic hydrolysate and its MRPs. (**A**): GC-IMS top view of the sample spectrum; (**B**): GC-IMS difference spectra of samples. “E” denotes the enzymatic hydrolysate; “M” denotes the MRPs.

**Figure 5 foods-11-04056-f005:**
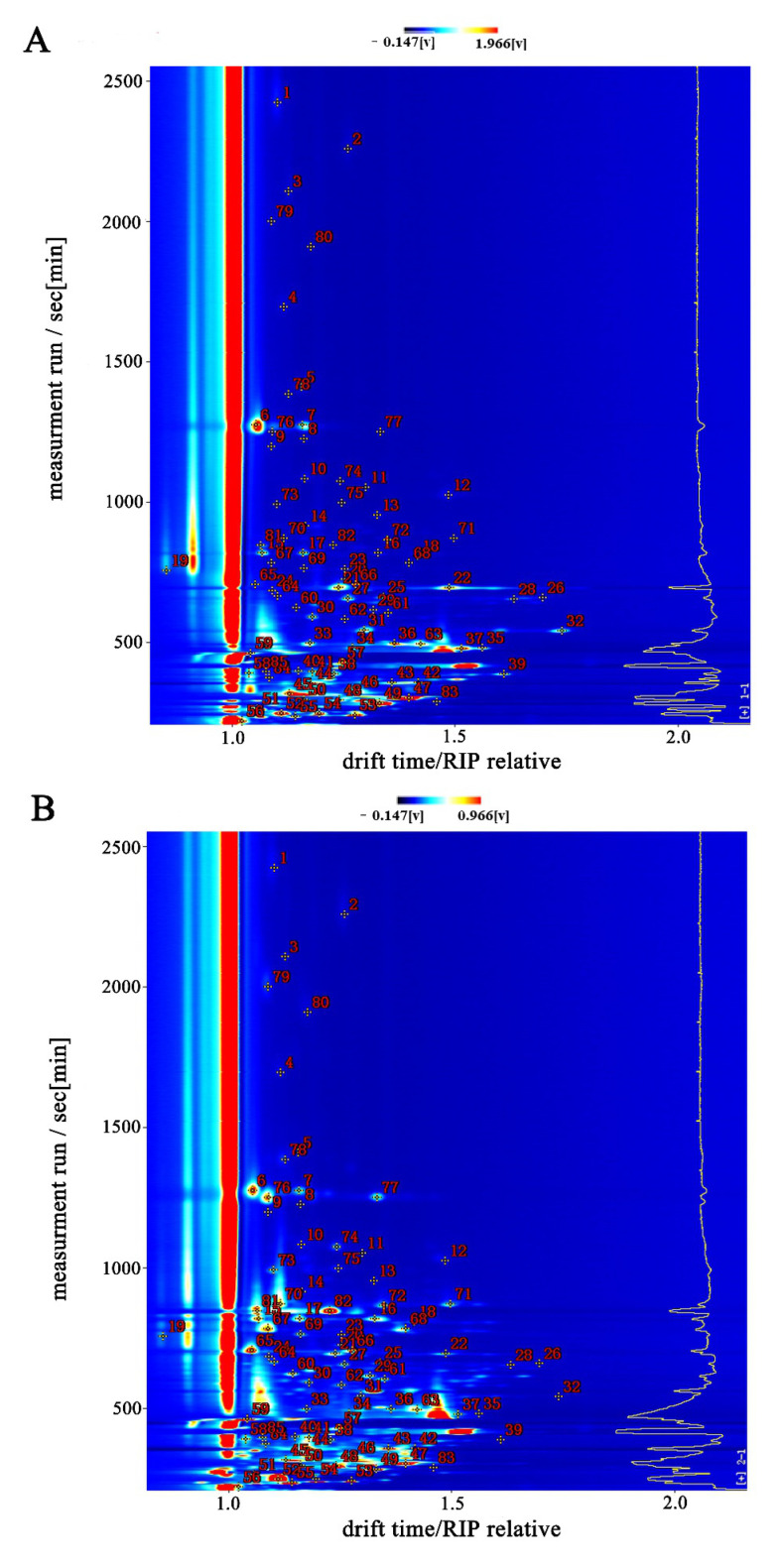
Topography of GC-IMS spectra of the enzymatic hydrolysate and its MRPs. (**A**): topography of GC-IMS spectra of the enzymatic hydrolysate; (**B**): topography of GC-IMS spectra of the MRPs. The numbers represent 84 known compounds.

**Figure 6 foods-11-04056-f006:**
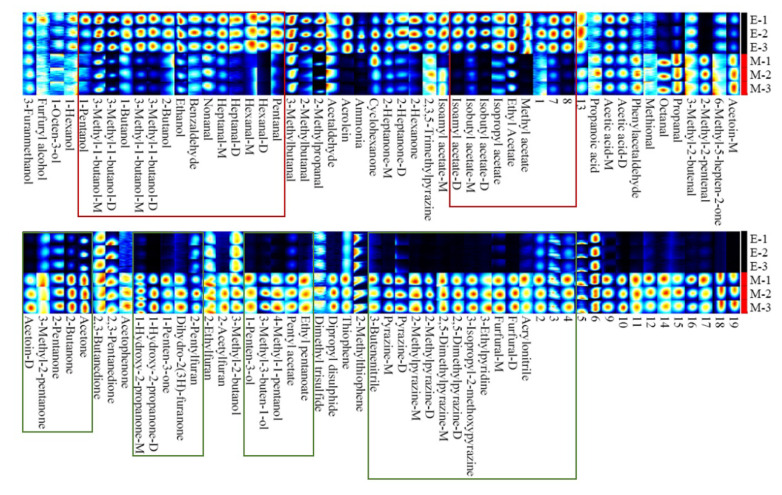
Gallery plot of the enzymatic hydrolysate and its MRPs. “E” denotes the enzymatic hydrolysatet; “M” denotes the MRPs. The numbers represent unidentified compounds. M and D after the compound indicate the monomer and dimer of the compound. The red box indicates that the concentration of the compound decreases after the MR, and the green box indicates that the concentration of the compound increases after the MR.

**Figure 7 foods-11-04056-f007:**
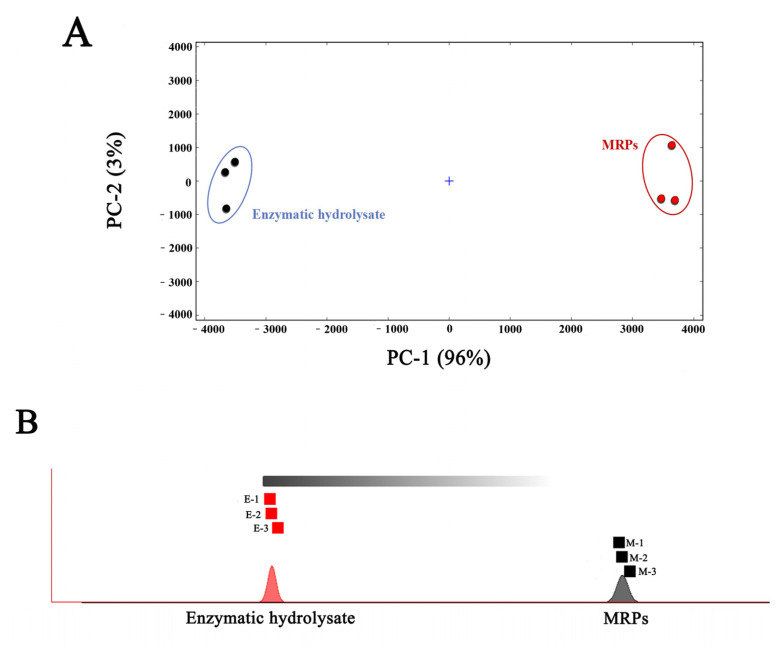
PCA plot (**A**), and nearest neighbor-Euclidean distance plot (**B**) of the enzymatic hydrolysate of *Lanmao asiatica* and its MRPs.

**Table 1 foods-11-04056-t001:** E-Nose sensor type.

Name	Type of Substance
W1C	Aromatic compounds
W5S	Nitrogen oxide
W3C	Ammonia, aromatic compounds
W6S	Hydride
W5C	Alkanes, aromatic compounds
W1S	Methane
W1W	Sulphides and terpenes
W2S	Alcohols, aldehydes, and ketones
W2W	Aromatic components and organic sulfides
W3S	Long-chain alkanes

**Table 2 foods-11-04056-t002:** GC-IMS integration parameters of volatile fractions in the enzymatic hydrolysatet of *L. asiatica* and its MRPs.

No.	Compound	CAS#	Formula	RI ^1^	Rt ^2^ [s]	Dt ^3^ [RIPrel]	Intensity (V)	
							E ^4^	M ^5^
	**Aldehydes**							
1	Phenylacetaldehyde	C122781	C_8_H_8_O	1764.8	2258.2	1.263	677.7	928.2
2	Benzaldehyde	C100527	C_7_H_6_O	1547.6	1409.4	1.158	553.8	321.5
3	Nonanal	C124196	C_9_H_18_O	1401.0	1025.3	1.487	501.5	492.8
4	Octanal	C124130	C_8_H_16_O	1291.0	807.9	1.412	133.1	627.5
5	3-Methyl-2-butenal	C107868	C_5_H_8_O	1211.9	685.8	1.093	151.6	149.4
6	Heptanal (M ^6^)	C111717	C_7_H_14_O	1194.7	661.8	1.342	723.7	356.7
7	Heptanal (D ^7^)	C111717	C_7_H_14_O	1194.2	661.2	1.699	132.3	41.7
8	Hexanal (M)	C66251	C_6_H_12_O	1097.6	481.2	1.273	420.7	167.5
9	Hexanal (D)	C66251	C_6_H_12_O	1097.6	481.2	1.564	863.6	101.9
10	Pentanal	C110623	C_5_H_10_O	998.7	358.7	1.420	906.3	314.7
11	3-Methylbutanal	C590863	C_5_H_10_O	926.2	304.1	1.399	9844.0	7841.7
12	2-Methylbutanal	C96173	C_5_H_10_O	916.5	297.6	1.623	307.6	220.5
13	Acrolein	C107028	C_3_H_4_O	862.0	263.2	1.059	7288.0	847.6
14	2-Methylpropanal	C78842	C_4_H_8_O	825.5	242.4	1.278	3199.6	596.4
15	Propanal	C123386	C_3_H_6_O	815.4	237.0	1.145	897.2	4572.6
16	Acetaldehyde	C75070	C_2_H_4_O	782.6	220.0	1.025	2310.7	1974.3
17	2-Methyl-2-pentenal	C623369	C_6_H_10_O	1176.2	624.5	1.147	54.1	1088.6
18	Furfural (M)	C98011	C_5_H_4_O_2_	1492.2	1249.7	1.093	713.4	4471.1
19	Furfural (D)	C98011	C_5_H_4_O_2_	1493.0	1251.8	1.336	78.5	1773.3
	Alcohols							
20	3-Furanmethanol	C4412913	C_5_H_6_O_2_	1797.3	2423.2	1.105	2459.8	2153.3
21	Furfuryl alcohol	C98000	C_5_H_6_O_2_	1732.9	2107.1	1.128	1052.3	891.3
22	1-Octen-3-ol	C3391864	C_8_H_16_O	1483.2	1225.6	1.163	126.1	109.9
23	1-Hexanol	C111273	C_6_H_14_O	1368.4	955.2	1.329	170.4	144.2
24	3-Methyl-1-butanol (M)	C123513	C_5_H_12_O	1218.4	695.2	1.241	4115.7	2607.9
25	3-Methyl-1-butanol (D)	C123513	C_5_H_12_O	1218.9	695.9	1.490	4122.6	1259.8
26	1-Pentanol	C71410	C_5_H_12_O	1263.1	762.6	1.256	140.0	107.6
27	1-Penten-3-ol	C616251	C_5_H_10_O	1166.0	603.8	1.353	50.3	455.0
28	3-Methyl-2-butanol	C598754	C_5_H_12_O	1106.2	495.0	1.426	1787.3	1527.1
29	3-Methyl-3-buten-1-ol	C763326	C_5_H_10_O	1263.7	763.5	1.164	120.2	993.7
30	4-Methyl-1-pentanol	C626891	C_6_H_14_O	1323.6	866.8	1.350	54.5	938.0
31	1-Butanol	C71363	C_4_H_10_O	1160.2	592.2	1.182	665.4	391.4
32	2-Methyl-1-propanol (M)	C78831	C_4_H_10_O	1108.1	498.2	1.178	1369.4	368.0
33	2-Methyl-1-propanol (D)	C78831	C_4_H_10_O	1107.6	497.4	1.367	1857.3	450.0
34	2-Butanol	C78922	C_4_H_10_O	1035.1	399.7	1.151	323.9	233.7
35	Ethanol	C64175	C_2_H_6_O	943.4	316.1	1.130	10,164.5	5460.7
36	Acetoin (M)	C513860	C_4_H_8_O_2_	1297.6	819.3	1.070	1054.7	2938.2
37	Acetoin (D)	C513860	C_4_H_8_O_2_	1297.3	818.6	1.330	148.6	2337.1
	Ketone							
38	6-Methyl-5-hepten-2-one	C110930	C_8_H_14_O	1348.6	915.2	1.167	218.0	169.3
39	Acetophenone	C98862	C_8_H_8_O	1687.7	1910.2	1.178	394.4	736.2
40	1-Hydroxy-2-propanone (M)	C116096	C_3_H_6_O_2_	1312.2	845.6	1.066	650.8	4227.2
41	1-Hydroxy-2-propanone (D)	C116096	C_3_H_6_O_2_	1313.4	847.9	1.229	130.1	10,203.7
42	Dihydro-2(3H)-furanone	C96480	C_4_H_6_O_2_	1708.8	1999.8	1.091	446.6	1694.8
43	1-Penten-3-one	C1629589	C_5_H_8_O	1031.1	395.0	1.079	37.4	217.3
44	Acetone	C67641	C_3_H_6_O	835.4	247.9	1.114	7043.1	22,187.3
45	2-Butanone	C78933	C_4_H_8_O	912.4	294.8	1.243	1304.9	7706.1
46	2-Pentanone	C107879	C_5_H_10_O	996.6	356.5	1.361	1295.1	2262.2
47	2,3-Butanedione	C431038	C_4_H_6_O_2_	993.4	353.8	1.179	825.0	981.8
48	3-Methyl-2-pentanone	C565617	C_6_H_12_O	1031.4	395.3	1.183	334.0	690.3
49	2-Hexanone	C591786	C_6_H_12_O	1096.8	479.9	1.518	3335.3	928.0
50	2-Heptanone (M)	C110430	C_7_H_14_O	1191.2	656.5	1.263	1156.3	558.7
51	2-Heptanone (D)	C110430	C_7_H_14_O	1190.9	655.8	1.635	344.5	126.9
52	Cyclohexanone	C108941	C_6_H_10_O	1296.9	817.9	1.162	1485.7	1397.8
53	2,3-Pentanedione	C600146	C_5_H_8_O_2_	1060.0	430.3	1.250	938.8	1218.5
	Esters							
54	Pentyl acetate	C628637	C_7_H_14_O_2_	1172.2	616.3	1.320	244.8	1025.2
55	Isoamyl acetate (M)	C123922	C_7_H_14_O_2_	1134.7	544.1	1.298	1794.2	1165.8
56	Isoamyl acetate (D)	C123922	C_7_H_14_O_2_	1133.9	542.7	1.742	1446.	109.0
57	Isobutyl acetate (M)	C110190	C_6_H_12_O_2_	1025.8	388.8	1.231	1321.3	474.4
58	Isobutyl acetate (D)	C110190	C_6_H_12_O_2_	1025.8	388.8	1.612	959.9	91.8
59	Ethyl Acetate	C141786	C_4_H_8_O_2_	894.8	283.4	1.333	13,691.2	5224.3
60	Methyl acetate	C79209	C_3_H_6_O_2_	833.5	246.8	1.198	2161.6	118.4
61	Ethyl pentanoate	C539822	C_7_H_14_O_2_	1155.6	583.2	1.255	57.8	247.7
62	Isopropyl acetate	C108214	C_5_H_10_O_2_	905.6	290.3	1.462	109.0	53.1
	Acids							
63	Propanoic acid	C79094	C_3_H_6_O_2_	1632.7	1695.1	1.118	554.0	556.5
64	Acetic acid (M)	C64197	C_2_H_4_O_2_	1501.3	1274.6	1.056	10,102.7	8938.7
65	Acetic acid (D)	C64197	C_2_H_4_O_2_	1501.3	1274.6	1.160	1870.8	1833.6
66	Methional	C3268493	C_4_H_8_OS	1472.7	1197.8	1.090	204.2	188.1
	Pyrazines							
67	3-Isopropyl-2-methoxypyrazine	C25773404	C_8_H_12_N_2_O	1423.2	1076.0	1.246	76.3	852.4
68	2,3,5-Trimethylpyrazine	C14667551	C_7_H_10_N_2_	1426.3	1083.2	1.166	351.4	268.9
69	2,5-Dimethylpyrazine (D)	C123320	C_6_H_8_N_2_	1326.1	871.5	1.499	50.1	1170.9
70	Pyrazine (M)	C290379	C_4_H_4_N_2_	1225.6	705.5	1.054	557.6	5283.2
71	Pyrazine (D)	C290379	C_4_H_4_N_2_	1226.1	706.3	1.279	272.2	1010.6
72	2-Methylpyrazine (M)	C109080	C_5_H_6_N_2_	1275.6	782.6	1.091	150.6	4035.6
73	2-Methylpyrazine (D)	C109080	C_5_H_6_N_2_	1275.6	782.6	1.399	100.2	2227.8
74	2,5-Dimethylpyrazine (M)	C123320	C_6_H_8_N_2_	1326.5	872.3	1.118	150.6	4035.6
	Furans							
75	2-Acetylfuran	C1192627	C_6_H_6_O_2_	1539.3	1384.1	1.129	185.5	641.15
76	2-Ethylfuran	C3208160	C_6_H_8_O	957.0	326.0	1.281	2234.0	2626.16
77	2-Pentylfuran	C3777693	C_9_H_14_O	1239.3	725.9	1.255	262.9	324.86
	Others							
78	Thiophene	C110021	C_4_H_4_S	1028.1	391.5	1.040	403.1	524.7
79	2-Methylthiophene	C554143	C_5_H_6_S	1084.3	462.5	1.044	4066.7	4764.9
80	3-Butenenitrile	C109751	C_4_H_5_N	1196.4	664.2	1.105	86.5	299.0
81	Acrylonitrile	C107131	C_3_H_3_N	1011.9	373.1	1.086	41.3	111.9
82	Ammonia	C7664417	H_3_N	1259.7	757.2	0.855	11,427.4	2980.7
83	3-Ethylpyridine	C536787	C_7_H_9_N	1385.4	991.2	1.103	46.2	650.0
84	Dipropyl disulphide	C629196	C_6_H_14_S_2_	1388.3	997.4	1.249	63.8	412.7

Note: ^1^: RI is retention index, ^2^: Rt is retention time, ^3^: Dt is migration time, ^4^: E is enzyme hydrolysate, ^5^: M is MRPs, ^6^: (M) indicates the monomer of the compound, ^7^: (D) indicates the dimer of the compound.

## Data Availability

Data is contained within the article.

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
