# Peer review of "Analysis of Volatile Flavor Substances in the Enzymatic Hydrolysate of Lanmaoa asiatica Mushroom and Its Maillard Reaction Products Based on E-Nose and GC-IMS"

_foods, 2022, doi:10.3390/foods11244056_

Round 1
Reviewer 1 Report
The presented article is devoted to the evaluation of the flavor of fungal hydrolysates using E-nose and gas chromatography-ion mobility spectrometry. The authors prepared the hydrolysate of Lanmao asiatica, then modified it with the Maillard reaction, and compared the flavors of these two samples. Principal component analysis was used for the statistical comparison of the samples. The research methodology corresponds to that used in chemometric flavor analysis and the raw material from which the hydrolyzate was obtained is used for the first time.
However, the disadvantage is the abundance of typos in the text, which needs to be significantly improved.
The following remarks need to be corrected:
- It is necessary to give the conditions for carrying out hydrolysis, namely the concentration of the substrate, the enzyme preparation, the pH of buffer, if it was used. It would be good to give the degree of hydrolysis of peptide bonds, so that it becomes clear whether the hydrolysis was deep or not.
- It is necessary to give the exact type of the gas chromatographic column (type of phase), but not just its sizes.
- The text lacks spaces between many phrases. There are combinations of several nouns that go in a row without any prepositions (bacterial bone enzyme hydrolysate, sensor differential contribution rate analysis, etc.). The description of the methods is given in a directive manner. It is necessary to improve the grammatical and stylistic quality of the text.
- The Figures 4a,b are too small and it is difficult to see what is shown there.
- There are no Dt and [RIP rel] in the column headings (Table 2). It is not necessary to give values up to 6 significant digits, since they were not defined with such precision.
Reviewer 2 Report
INTORDUCTION
1. Bitterness is taste, not flavour
2. Please explain the possible mechanizes, of Mailliard reaction in this mushroom. What kind of amino acid did involved? Content of reducing sugar? Processing and temperature? etc.
3. What do you mean with "increase the freshness"?
4. Please give explanation regarding the importance of combining E-Nose and GC-MS? If GC-MS is more powerful to obtain data, why did the author also use E-Nose? Please explain the advantages and disadvantages of each method, and thus give rationale on combining the analysis methods.
MATERIAL AND METHOD
1. Please change "Materials:..." to "Materials used in this study included..."
2. Age and other specification of the mushroom must be explained.
3. Please change "Equipment:..." to "Equipment used in this study included..."
4. This study will be better if includes the control sample (without enzymatic treatment and Maillard reaction).
5 "The enzymatic hydrolysate was carried...." to "The enzymatic hydrolysate production was carried..." ???
6. What kind of enzyme did involved?
7. Enzym inactivation by using what? Waterbath? Oven?
8. Some statements are difficult to understand. Going through pofessional proofreader is highly recommended.
9. Section 2.2.2.: Decimal on 10.00% is not necessary
10. Section 2.2.3: Is the E-nose coupled with polar and polar column? Please explain.
11. Section 2.2.3: What do you means with samples? Hydrolysate? MRPs? PCA generated from E-nose analysis was based on the discrimination test. Therefore, there must be more than 1 sample. Please explain in a more detail.
12. Section 2.3: Please include the explanation of data analysis in the sub-sections of E-Nose analysis and GC-IMS analysis (depending on the content).
RESULT AND DISCUSSION
1. Fig 3: Each sample has 12 repetition? Please explain in the section of method.
2. Is Table 2 really important? What do E and M stand for? Is the result comparable with E-nose?
CONCLUSION
1. Please explain the limitation of this study, and recommendation for further research.